# Improving work detection by segmentation heuristics pre-training on factory operations video

**Shotaro Kataoka** [1]*, **Tetsuro Ito**[2], **Genki Iwaka**[2], **Masashi Oba**[2], **Hirofumi Nonaka**[2]

**1** Department of Science of Technology Innovation, Nagaoka University of Technology, Nagaoka, Niigata, Japan, **2** Department of Information Management Systems Engineering, Nagaoka University of Technology, Nagaoka, Niigata, Japan

* s183347@stn.nagaokaut.ac.jp

**Data Availability Statement:** All video and annotation files are available from https://www.kaggle.com/datasets/shotarokataoka/factory-oparation-video-dataset.

## Abstract

The measurement of work time for individual tasks by using video has made a significant contribution to a framework for productivity improvement such as value stream mapping (VSM). In the past, the work time has been often measured manually, but this process is quite costly and labor-intensive. For these reasons, automation of work analysis at the work-site is needed. There are two main methods for computing spatio-temporal information: by 3D-CNN, and by temporal computation using LSTM after feature extraction in the spatial domain by 2D-CNN. These methods has high computational cost but high model representational power, and the latter has low computational cost but relatively low model representational power. In the manufacturing industry, the use of local computers to make inferences is often required for practicality and confidentiality reasons, necessitating a low computational cost, and so the latter, a lightweight model, needs to have improved performance. Therefore, in this paper, we propose a method that pre-trains the image encoder module of a work detection model using an image segmentation model. This is based on the CNN-LSTM structure, which separates spatial and temporal computation and enables us to include heuristics such as workers' body parts and work tools in the CNN module. Experimental results demonstrate that our pre-training method reduces over-fitting and provides a greater improvement in detection performance than pre-training on ImageNet.

## 1 Introduction

Recognition of employee activities in the production process is gaining attention [1–12]. Bell developed a conceptual framework about artificial intelligence data-driven Internet of Things systems [1]. Elvira et al carried out to evaluate and analyze artificial intelligence-supported workplace decision method such as big data algorithmic analytics, sensory and tracking technologies [2]. Ren et al carried out a comprehensive review of big data analytics throughout product lifecycle to support sustainable smart manufacturing [3]. Smith analyzes the outcomes of cyber-physical manufacturing systems such as real-time sensor networks, and Internet of

**Funding:** We received a Research Grant from Nagaoka University of Technology, to which we belong. Kataoka and Nonaka are currently receiving salary from Nagaoka University of Technology. Inagaki and Iwaka had received salary from Nagaoka University of Technology. The funders had no role in study design, data collection and analysis, decision to publish, or preparation of the manuscript.

**Competing interests:** NO authors have competing interests.

Things smart devices [4]. Clarke draw on a substantial body of theoretical and empirical research on big data-driven manufacturing, and to explore this, I inspected, used, and replicated survey data from Accenture, Capgemini, PwC, Software AG, and we.CONECT, performing analyses and making estimates regarding sensing, smart, and sustainable technologies [5]. Nica et al performed analyses and made estimates regarding the link between smart connected sensors, industrial big data, and real-time process monitoring to examine cyber-physical system-based manufacturing [6]. Leng et al presents a digital twin-driven manufacturing cyber-physical system (MCPS) for parallel controlling of smart workshop under mass individualization paradigm [7]. Hyers et al. analyzed big data-driven decision-making processes, Industry 4.0 wireless networks, and digitized mass production to evaluate utility of cyber-physical system of smart factories [8]. Keane et al performed structural equation modeling to analyse cognitive automation, big data-driven manufacturing, and sustainable industrial value creation using data collected from BCG, CompTIA, Deloitte, IW Custom Research, Kronos, McKinsey, PAC, PwC, and Software AG [9]. Mircica et al. analyzed and estimated regarding the main areas of interest for manufacturers within Industry 4.0 and the effects of Industry 4.0 on the workforce [10]. Graessley et al performed analyses and making estimates regarding how Industry 4.0 is delivering revenue, cost and efficiency gains and top technologies being considered in-line with organizations' strategic plan [11]. Meyers collected data from Bright & Company, Corporate Research Forum, Deloitte, Management Events, McKinsey, and Top Employers Institute and estimated data-driven approaches in industry [12]. Among them, employee behavior recognition in the manufacturing industry is attracting increasing attention from both industry and academia [13]. For example, to improve management and effectiveness of employees' learning processes, Patalas-Maliszewska et al. [14] used CNN-SVM to extract features from video material concerning each work activity, and comparison with the features of the instruction picture. The importance of real-time process monitoring and analysis of employee behavior based on various sensors and other devices has been increasing from both industry and academia [15–18]. In such a situation, the measurement of work time for individual tasks by using video and other methods has made a significant contribution to a framework for productivity improvement such as value stream mapping (VSM) [19] which is a method for reviewing and improving productivity by dividing tasks into two categories—value-adding tasks and non-value-adding tasks—and by analyzing the time spent on the tasks [20–24]. Monteiro et al performed case study analysis of lean tool. In the case of metalworking company project, flowcharts and VSM (Value Stream Mapping) approaches were conducted to identify and map key processes and improved setup times. As a result, setup times were reduced in 40% on the vertical milling machine of the company, and in 57% on the horizontal milling machine [20]. UlfKTeichgraber et al. have empirically demonstrated that VSM can be applied to identify NVAs and improve productivity in the manufacturing process of endovascular stents. Specifically, they found that 5 out of 15 processes were unnecessary and could be eliminated by using VSM [21]. Mareike Heinzen et al. clarified the effect of co-location in the new drug development process through VSM. The results suggest that communication increases especially in child-located teams [22]. Heravi et al. used value stream mapping (VSM) as a lean method in the manufacturing process of prefabricated steel frames (PSF) and showed that 34% of the lead time could be reduced [23]. Wang et al. developed a training method combining VR and VSM, and showed that it can contribute to the efficiency of the construction process [24]. It has made a significant contribution to improve work efficiency and has been successfully introduced to a variety of industries [25–28]. SMED provides the most complete and detailed procedures available anywhere to transform the manufacturing environment in a way that speeds production and makes small lot inventory feasible [25]. Zhang et al. found that lean manufacturing contributes to knowledge creation through a

questionnaire survey [26]. Sousa et al. applied VSM to improve production efficiency in the cork industry, and found that production time could be reduced by 43% [27]. Adanna et al. showed that VSM can reduce set-up time by 45% when applied to automotive equipment manufacturing [28]. An important part of implementing such a VSM is to measure work times in order to visualize them. For this purpose, video analysis is often used. For example, [29] used video to extract the cycle time and other factors that are important in analyzing the value added by a task. However, this type of work time analysis requires experts who are well versed in the work process and also requires considerable labor and involves economic costs because of the need to check the video footage during the work. Thus, the implementation hurdles are high, especially in small and medium-sized companies. For these reasons, automation of work analysis at the worksite is needed. In the mainstream research on this topic, sensors attached to the operator have traditionally been used for collecting measurement data. However, this method has certain problems, such as the burden of attaching a sensor to the worker, the resulting limitation on the worker's flexibility of movement, and the high cost of introduction. Another approach is to detect action from videos; among such methods, convolutional neural network (CNN)–based methods are the most popular. There are two main methods for computing spatio-temporal information: by 3D-CNN, and by temporal computation using LSTM after feature extraction in the spatial domain by 2D-CNN. The former has high computational cost but high model representational power, and the latter has low computational cost but relatively low model representational power. In the manufacturing industry, the use of local computers to make inferences is often required for practicality and confidentiality reasons, necessitating a low computational cost, and so the latter, a lightweight model, needs to have improved performance. In addition, both methods tend to overlearn when the training dataset is small, and it is difficult to prepare a large dataset with a large number of work annotations for actual applications in the manufacturing industry. Therefore, the automation of work time analysis is a highly important topic of research. However, conventional video analysis and sensor-based methods are difficult to apply to the analysis of work hours in the manufacturing industry in environments where the granularity of work is not uniform and workers can move around freely. This has posed a major problem, especially in applying the system to manufacturing sites with high-mix low-volume production. In this paper, therefore, we propose a method using machine learning to train a model to extract the features of a worker's position and posture from a set of video clips taken in a factory and to then use the model to detect the content of the worker's work.

Our contributions can be summarized as follows:

1. We propose the printing factory video dataset, which is a dataset of videos taken from a third person's point of view using a fixed camera and containing second-by-second work content annotations. The dataset is available at https://www.kaggle.com/datasets/shotarokataoka/factory-oparation-video-dataset.

2. We propose a method for considering heuristics in a model that uses an image segmentation task as a pre-training method. To our best knowledge, there is no research on the application of such a method to work videos in factories. In particular, the video proposed in this study has completely new characteristics, as described in Table 2, and it would be of great academic value to confirm the effectiveness of heuristic pre-training on this video.

3. We present our findings that with our dataset, the proposed pre-training method significantly reduced over-fitting and improved the F1 score on the test set by 0.2906 points over the ImageNet pre-training method. The code and raw result are available at https://github.com/ShotaroKataoka/WorkDetectionFactoryPretrain.

The structure of this paper is as follows. In section 3, we review related works. Then, in Section 4 we define the task. In section 4, we explain about proposal method. In section 5, we explain about experiment. In section 6, we showed the experiment result. Finally, in Section 5. we dicussed abou the result. The paper finishes with some conclusions and points out some future work.

## 2 Related work

In order to automate the work process analysis, it is necessary to estimate the work content and the time it took to perform it from some data. In this paper, we refer to this sequence of steps as "work detection." There are two approaches to work detection: using a sensor (e.g., an accelerometer) to collect data on the physical movement of a worker and using a camera to record videos of the work. In this section, we describe previous studies on sensor-based and video-based task estimation, video datasets, and methods of action recognition.

### 2.1 Action recognition from sensors vs. videos

With the development of sensing devices, research on motion recognition using sensor information has been attracting attention [30–50]. Peterek compared different algorithms for human physical activity recognition from accelerometric and gyroscopic data which are recorded by a smartphone [30]. Chang et al. proposed a hierarchical hand motion recognition method based on one inertial measurement unit (IMU) sensor and two surface electromyogram (sEMG) sensors. An SVM classifier was used for the EMG signal and a decision tree classifier was used for the IMU signal [31]. Ronao developed a two-stage continuous hidden Markov model (CHMM) approach for the task of activity recognition using accelerometer and gyroscope sensory data gathered from a smartphone [32]. Uddin et al. propose a model that extracts feature vectors from multi-sensor data such as ECG based on Gaussian kernel-based principal component analysis and Z-score normalization, and then trains them with CNN [33]. Wang et al. also use CNN to analyze human activity [34]. Lee et al. have developed a method to analyze time-series acceleration signals using 3D acceleration sensors of smart phones using a hierarchical hidden Markov model [35]. Ravi et al. found that using Plurality Voting showed good motion recognition performance even with a single accelerometer [36]. Kwapisz also used a single accelerometer to recognize human activities [37]. Casanova et al. have shown that recognition of motion features from acceleration data can be applied to biometric authentication [38]. Albinali proposed a motion recognition algorithm for individuals with the Autism Spectrum Disorder (ASD) by combining the Fourier transform of acceleration time series with decision trees [39]. Khan developed a model that combines an artificial neural network (ANN) and an autoregressive model (AR) to recognize motion from acceleration data [40]. Kaghyan et al. proposed an algorithm that uses accelerometer data and the K-NN algorithm to identify common activities performed by the user [41]. Brezmes also used K-NN to recognize human activities [42]. Mitchell et al. have developed a method for sports motion recognition from accelerometers using Discrete Wavelet Transform (DWT) and SVM [43]. Subasi et al. showed that an ensemble classifier based on the Adaboost algorithm using acceleration data can significantly improve the performance of automatic human activity recognition (HAR) [44]. Wang et al. proposed a real-time, hierarchical model to recognize both simple gestures and complex activities using a wireless body sensor network [45]. Garcia-Ceja used hidden Markov models and conditional probability fields to recognize actions based on time series of sensor information [46]. Hossain et al. proposed a method that combines accelerometers with LoRaWAN sensors to recognize basic actions such as walking, staying, and running by KNN and LDA [47]. Ryu et al. showed that accelerometer information can be used

**Table 1. Comparison of sensors and video.**

| Device | Movement flexibility | Cost | Representation |
|---|---|---|---|
| Sensors | × | × | × |
| Video | o | o | o |

to classify a simple routine task consisting of four subtasks using a multi-class SVM [48]. Kim et al. analyzed IMU data by dynamic time warping (DTW) to classify the movements of construction machines [49]. These studies are of general action recognition. Sensors have been used in a number of studies, such as in an analysis of work motions in bicycle maintenance using ultrasonic and IMU sensors [51], an analysis of line motions using only IMU sensors [52], the estimation of hammering and other motions on an assembly line using wrist-mounted IMU sensors [53], and the estimation of work types using wristwatch IMU sensors [54, 55]. Modern sensor-based work recognition methods that use deep learning include those proposed by [56, 57], which use a convolutional neural network (CNN) to analyze data from IMU sensors.

Such methods use IMU sensors and other wearable sensors primarily to sense the movement of a specific body part such as the hand and are intended for the analysis of assembly work in which the worker is stationed at a workbench and is not required to walk around. However, it is difficult to apply a sensor-based method to an environment in which workers can move freely because 1) parts of the body that are not equipped with sensors can move, 2) the direction of movement ranges widely, and 3) the body position must be accurately estimated. To capture workers' movements in such high-flexibility environments, expensive devices such as multi-purpose, infrared, and line-of-sight sensors may be required. Under such conditions, it is difficult to sense the work as performed in a natural state because of the burden of mounting the sensor. Therefore, the application of sensor-based work recognition methods has been limited to the analysis of production lines in which only specific tasks with fixed actions are performed.

Unlike sensors, video is a rich source of information and can extract the features of the work even if the degree of flexibility in the worker's movement is high; in addition, locations can be accurately identified. Furthermore, the use of a small camera has the advantage of facilitating the construction of an inexpensive system.

Table 1 summarizes the characteristics and capabilities of sensors and videos for action recognition.

As described in Section 3 below, in a printing factory such as the one used for the work analysis in this study, the worker has a high degree of movement flexibility; therefore, gathering data by using a sensor was not considered suitable from the standpoint of the burden on the worker. Therefore, in this study, we adopted a video-based work detection method.

## 2.2 Video datasets used for action recognition

Recently, there have been many studies on video-based general action recognition, focusing primarily on large public video datasets created through crowdsourcing. The characteristics of datasets commonly used in action recognition/detection research are listed in Table 2. UCF101 [63], Kinetics [59, 60], Something-Something [62] HMDB51 [79], and Sports-1M [65] are particularly well-known. However, they are used for action recognition tasks rather than for action detection because unlike our proposed dataset, each of their videos contains only one action. "AVA" is a movie action dataset, whichconsists of multi-label annotations of basic actions such as "talk to" and "watch" [66]. "NTU RGB+D" contains 60 classes of human

**Table 2. Comparison of video datasets.**

| Dataset name | Fixed camera? | Multi-label&time detection? | Action/Work | Content | Time range |
|---|---|---|---|---|---|
| Charades [58] | no | yes | general work | general | seconds |
| Kinetics [59, 60] | no | no | action | general | seconds |
| EPIC-Kitchen [61] | no | yes | specific work | cooking | minutes |
| Something-Something [62] | no | no | action | general | seconds |
| UCF101 [63] | no | no | action | general | seconds |
| HMDB51 [64] | no | no | action | general | seconds |
| Sports-1M [65] | no | no | action | general | seconds |
| AVA [66] | no | no | action | general | seconds |
| Ntu rgb+ d [67] | no | no | action | general | seconds |
| Egogesture [68] | no | no | action | general | seconds |
| THUMOS14 [69] | no | no | action | general | seconds |
| Activitynet [70] | no | no | action | general | seconds |
| Volleyball [71] | no | no | action | general | seconds |
| Win-Fail [72] | no | no | action | general | seconds |
| HAA500 [73] | no | no | action | general | seconds |
| Youtube-8M [74] | no | no | action | general | seconds |
| Hollywood in Homes [58] | no | no | action | general | seconds |
| Moments in Times [75] | no | no | action | general | seconds |
| Hacks Clip [76] | no | no | action | general | seconds |
| HVU [77] | no | no | action | general | seconds |
| AViD [78] | no | no | action | general | seconds |
| Ours | yes | yes | specific work | factory | seconds to minutes |

activitivity and 56,880 video samples [67]. The labels in these two datasets are in three major categories: daily actions, mutual actions, and medical conditions. These labels consist of basic actions or states such as "drink water" or "staggering" [67]. "EgoGesture" is a multi-modal large scale dataset for egocentric hand gesture recognition. This dataset has 83 classes of static or dynamic gestures such as "OK" [68]. "THUMOS Challenge 2014 Temporal Action Detection" dataset defines specific motions in sports, such as dunking and pitching [69]. "Activity Net" Temporal Action Localization is a dataset for action recognition consisted of 200 different daily activities such as: "walking the dog", "long jump" [70]. Ibrahim et al. collected a new dataset using publicly available YouTube volleyball videos. They annotated 4830 frames that were handpicked from 55 videos with 9 player's basic action labels such as "spiking" [71]. "Win-Fail Action Recognition" is a task for judging the success or failure of an action in the following domain: "General Stunts," "Internet Wins-Fails," "Trick Shots," & "Party Games." [72]. "HAA500" contains fine-grained atomic actions where only consistent actions are classified under the same label (e.g., "baseball pitching" and "basketball free throws" to minimize ambiguity in action classification) [73]. "YouTube-8M" is the largest multi-label video classification dataset, consisting of up to 8 million videos (500,000 hours of video), annotated with a vocabulary of 4800 visual entities [74]. The Moments in Time Dataset is a large human-annotated collection of one million short videos corresponding to dynamic events that unfold in less than three seconds [75]. HACS is a dataset for human action recognition using a taxonomy of 200 action classes, which is identical to that of the ActivityNet-v1.3 dataset [76]. It has 504K videos retrieved from YouTube and each one is strictly shorter than 4 minutes, and the average length is 2.6 minutes. A large-scale "Holistic Video Understanding Dataset" (HVU) is organized hierarchically in a semantic taxonomy that focuses on multi-label and multi-task

video understanding [77]. AViD is a large-scale video dataset which has 467k videos and 887 basic action classes. The AViD dataset consists of similiar actions to those in Kinetics, plus some additional actions such as talking, explosion, boating [78]. HDMB is the largest action video database with 51 basical action categories, which in total contain around 7,000 manually annotated clips extracted from a variety of sources ranging from digitized movies to YouTube [79]. "something something" is a large collection of labeled video clips that show humans performing pre-defined basic 174 actions with everyday objects [80]. "Charades" is composed of 9,848 annotated videos with an average length of 30 seconds involving 157 action classes such as "pouring into cup" [58]. "Kinetics-600" is a large-scale action recognition dataset which consists of around 480K videos from 600 action categories [58]. Each video in the dataset is a 10-second clip of action moment annotated from raw YouTube video [81]. "Kinetic-700" is an extensions of the Kinetics-600 dataset [82].

Charades [58] and EPIC-Kitchen [61] contain multiple actions per video, and, like our dataset, they are intended for action detection tasks. Charades is a dataset of videos consisting of common actions such as "Holding a laptop" or "Watching TV." Each of these actions ranges from a few seconds to tens of seconds in length; the length of time required for each action is small. EPIC-Kitchen is a video dataset with clips of first-person work activities in the kitchen; it differs from general action datasets such as Charades in that it consists of complex work activities in the specialized domain of cooking. In addition, the duration of each work activity is on the order of minutes, and it includes a wide range of work durations, although not as wide as our dataset.

Given these characteristics, EPIC-Kitchen is the dataset most similar to ours. There are three important differences, however, between our dataset and EPIC-Kitchen. First, our dataset covers work in the factory. To the best of our knowledge, there is no other video dataset for work detection in the factory. Therefore, it has been difficult to automate the operation process analysis using video; we believe that the proposed dataset will contribute to the progress of future research. In addition, unlike conventional general action datasets, this dataset is intended for the analysis of specialized work activities consisting of series of complex body movements. Second, our dataset includes a wider range of time periods than EPIC-Kitchen, from a few seconds to several tens of minutes per work activity. In general, having a wider range of work times increases the difficulty of work detection because the model needs to capture both micro and macro changes in the video. As mentioned above, the range of work times has been limited in the past, and therefore models that address this problem have not been well studied. We believe that this dataset will contribute to the development of research on work detection over long periods. Third, the videos in our dataset were captured by a fixed-point camera. Considering actual application to the work site, the method whereby a camera is affixed to the operator and acquires video, such as was done for the videos in EPIC-Kitchen, is not appropriate from the standpoint of the burden on the operator; it is also costly because it is necessary to prepare a camera for each worker. In addition, it has been pointed out that in cases where the background information varies, the work content may be estimated using only the features of the background [83]. An example that is mentioned is that if the background is a tennis court, it is possible to infer that the action in the video is "Tennis Swing" even if the person is removed from the video. If a first-person camera is used for task detection as in EPIC-Kitchen, we can expect that each work activity will have its own characteristic background (e.g., a sink when the activity is washing dishes). Because our dataset uses a fixed-point camera, the background is the same throughout the video. Therefore, it is possible to avoid the problems pointed out by [83] and set up a task that will infer the work content from the worker's movements rather than the background.

**Table 3. Comparison of 3D-CNN and CNN-LSTM.**

| Model type | Segmentation? | Inference memory | Time range |
|---|---|---|---|
| 3D-CNN | no | large | short |
| CNN-LSTM | yes | small | long |

## 2.3 Action recognition methods

Until the use of CNNs became widespread, manually defined features were proposed as the analysis model for use with the above action recognition datasets. Examples include Oreifej and Liu et al.'s HOND4 feature [84], the HOG feature-based model [85], STIP [86], Gradient and Histogram of Optical Flow [87], 3D Histogram of Gradient [88], SIFT-3D [89], and SURF [90]. The method using Fisher vector representation [91] was also mainstream.

Since the publication of the study by [92], CNN methods have become mainstream in the field of computer vision, and in the video processing field, the SOTA method is a model based on CNN. Video processing requires not only spatial computation, as in the case of image processing, but also temporal computation. There are two main methods for computing the temporal dimension in CNN-based methods (see Table 3 for a summarized comparison). The first is a 3D convolutional neural network (3D-CNN) such as C3D [93]. This is a method for computing both spatial and temporal convolutions simultaneously; it has high representational capability. Following the proposal of C3D, 3D-CNN has been actively studied, such as in I3D [60], which employs Inception, and in work by [94], which is based on ResNet. Qui et al. proposed the Local and Global Diffusion (LGD) model. This model is equipped with LGD blocks that learn local and global representations in parallel [95]. D3D is a method for spatio-temporal recognition by distillation. Specifically, it is based on a teacher network that grasps the features of the optical flow and learns from the features of RGB only with a student network [96]. Tran et al. devised a model consisting of 3D ResNet by searching for the optimal architecture of convolutional neural networks [97]. PERF-Net, which uses the rendering information of poses in RGB frames to capture the motion features, has also been proposed [98]. In addition to the two streams, Hong et al. propose to use additional pose and pairwise streams which can be extracted contextual action cues to improve the performance of action recognition [99]. Shuiwang et al. have proposed a method for action recognition using 3D convolutional neural networks [100]. FstCN [101] also decompose a 3-dimensional convolution into 2-dimensional and 1-dimensional convolutions the model to reduce the amount of computation. ResNet [102] and DenseNet [103] has also been proposed as a model to reduce computational complexity. X3D is a model that searches for the optimal network structure [104]. A method to reduce the number of channels, called R(2+1)D [105], has been proposed, and the computation efficiency is being improved. Many other methods based on 3D-CNN such as P3D [106], S3D [107], and CSN [108] have been proposed and are SOTA for Kinetics and other datasets [109].

The second method of CNN-based video processing is one that first calculates spatial directions with an ordinary 2D-CNN and extracts features of the frame and then calculates temporal direction with LSTM or CNN [110]. This method is inferior to the 3D-CNN method in terms of representational capability because the spatial and temporal calculations are separate from each other, but it is advantageous in terms of computational cost. In the case of a large model such as 3D-CNN, the computer specifications are too high to introduce it to a factory, and it might not be possible to infer the actual environment. Therefore, there is a strong need to develop a relatively lightweight model such as CNN-LSTM. In addition, in order to create a work detection model for operation process analysis, it is necessary to create a video dataset

with work annotations for each work site. As the cost of creating such a dataset is very high, it is difficult to create a large dataset. With consideration of this background, we investigated ways of creating a better model by training a conventional basic CNN-LSTM model on a small dataset.

## 3 Task definition

In this study, we wished to analyze the work time of the main worker by categorizing his or her work content by the second from video recordings of factory work. The particular focus was to develop a method of factory work analysis that could be used in real-world applications. However, we did not find any video datasets available in the market that met the requirements for video content and labeling. Therefore, a dataset was created for this study by recording three days' work content in an actual factory using a common video camera. Fig 1 shows an example of one frame of video recorded for the dataset. The task was then to classify the content of the main worker's work by the second to annotate this dataset.

The proposed work video dataset has several features that differ from public video datasets that are conventionally used:

- The videos are captured by a fixed camera from a bird's eye view in an actual printing factory.

- The videos are untrimmed and have a length of about 10 h per day. There is approximately 30 h of labeled video for the entire dataset.

- Videos show the full body of the worker and include complex movements. The background environment may also vary independently of the work.

- The worker uses multiple machine tools and small tools to perform the work.

- Twelve work labels were defined to categorize the work content. (These are not the same as action labels. Therefore, even if two physical actions are similar, they will be assigned different work labels if they differ in terms of work content.) With consideration of this background, we investigated ways of creating a better model by training a conventional basic CNN-LSTM model on a small dataset.

- The length of time to perform the same work consecutively varies considerably, ranging from a few seconds to tens of minutes, and tasks depend on one other over a very long period.

## 4 Method

The data were collected after prior explanation to the company (Echigo Fudagami Inc) that the data would be used for research purposes and verbal consent was obtained from the company manager and the subject workers. Permission to conduct the research was obtained from the company manager and subject workers. Our university's ethics committee confirmed that ethical approval was waived. And no third-party ethical oversight was provided.

In this section, we first describe the work detection model used in our method, that is, the series of processes for classifying the operator's work by the second based on a series of input images, as shown in Fig 2.

Then, we explain our approach for improving the generalization performance by pre-training this model on a task to segment workers. Even with pre-training on the segmentation task, the model structure in the inference process is almost identical to that of the work detection

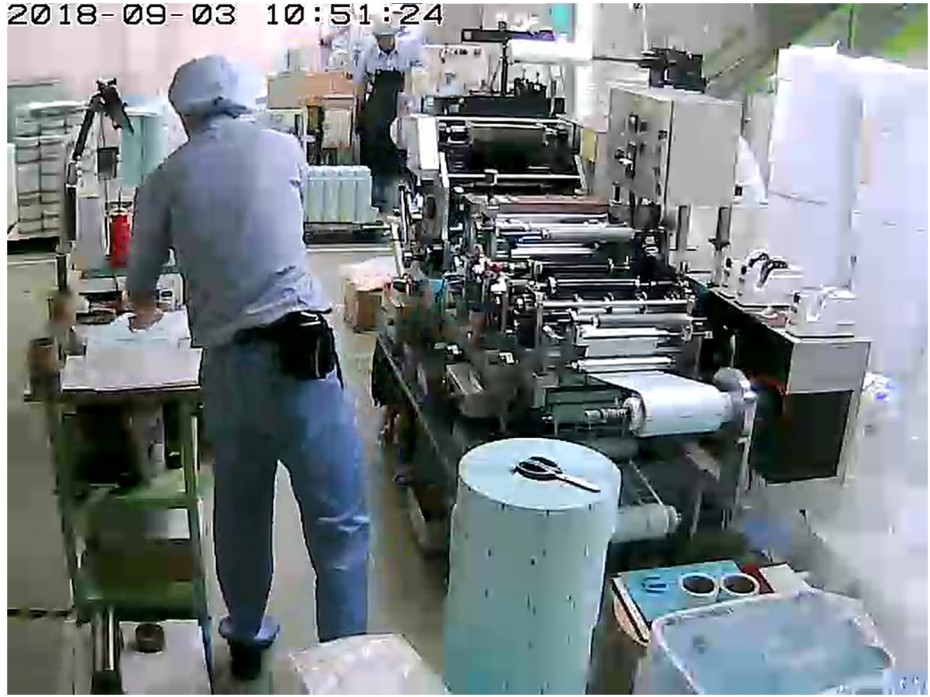

**Fig 1. Sample frame of proposed factory video dataset.**

model described above. Therefore, we expected the pre-training to improve performance considerably without increasing the computational cost of inference.

For the pre-training, an encoder–decoder segmentation model [111–113] is used; this makes it possible for the model to effectively learn important information such as the body parts of the worker from a small quantity of data. During the training of the work detection model, only the encoder module is used, to enable the feature extraction to focus on the important information.

The dataset videos were filmed using a fixed camera for three days at a specific work site in a printing factory (Echigo Fudagami Inc: Uenoyama 1-2-8, Ojiya, Niigata, Japan.), with a data size of 10 h per day, i.e., 30 h for the entire dataset.

## 4.1 CNN-LSTM model

The CNN-LSTM model for work detection is shown in Fig 2. The algorithm of this step is explained in Algorithm 1. In the hyper-parameters for each layer in Fig 2, the default values described in Fig 3 are used unless a value is specified. The hyper-parameters of the LSTM were determined through grid search by comparing validation scores. Each frame image of the video data is entered into the same CNN model [92]; the CNN model computes the images in a convolutional calculation and converts them into feature vectors. Then, a series of these feature vectors are input to LSTM [114]. LSTM performs feature extraction by considering the temporal patterns of the feature vectors of the frames. After that, the fully connected layer and Softmax infer the work content for each frame.

For video-based action detection, let $X_n = \{x_n, \ldots, x_{n+T}\}$ denote a series of frames starting from time $n$, where $T$ is the length of the input sequence.

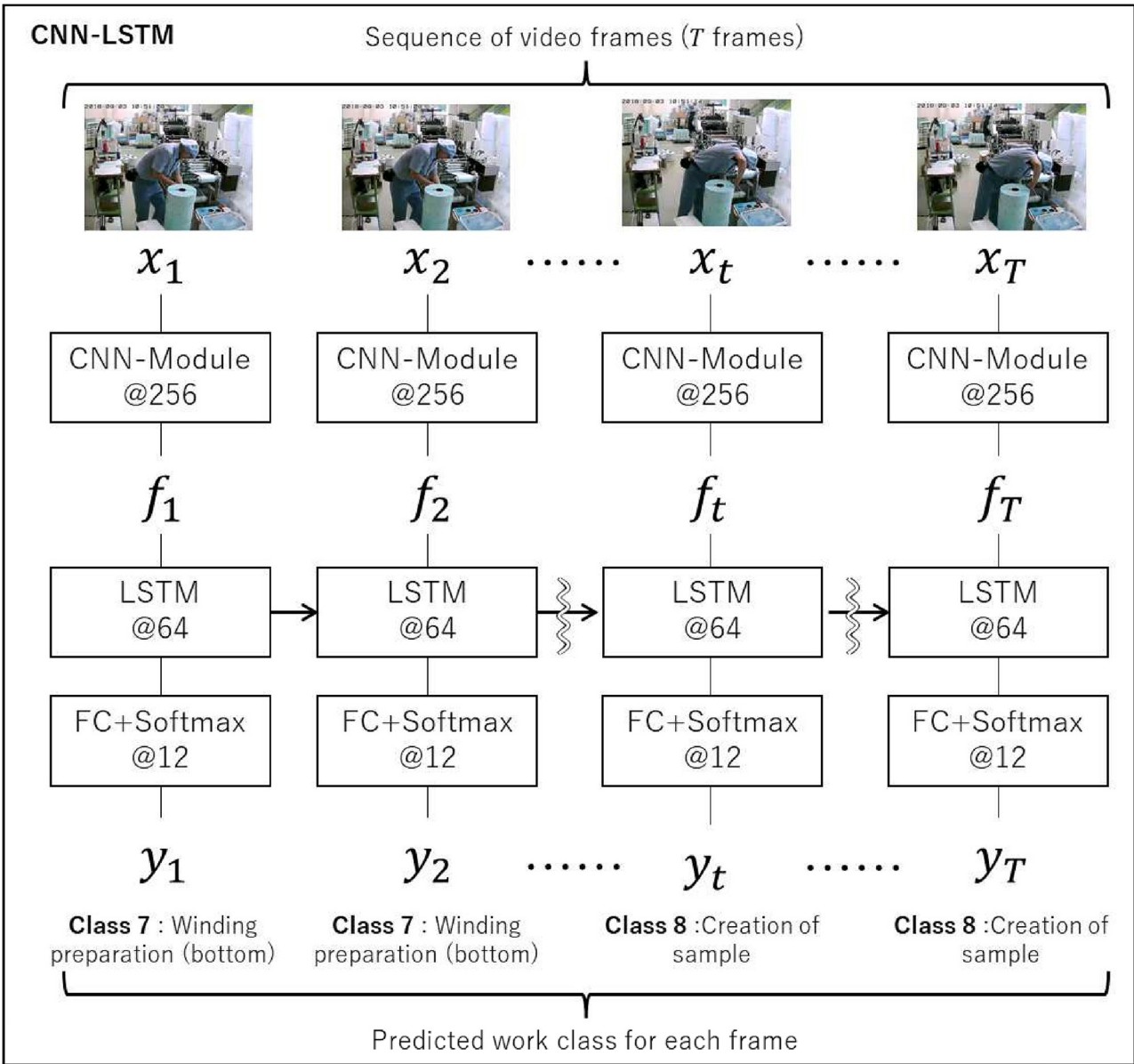

**Fig 2. CNN-LSTM model architecture (baseline and ours).** All CNN-Modules and FCs share weights with each other. The number marked @ is the output channel. Each layer component corresponds to the class diagram in Fig 3. Hyper-parameters not listed use the default values from Fig 3. CNN-Module: It is described later in Fig 4.

The CNN-Module performs the following calculation:

$$f_t = CNN - Module(x_t) \tag{1}$$

where $x_t$ is the $t$-th frame of $X_n$, and *CNN-Module*($\cdot$) is CNN feature extractor that transforms a 2D image into a vector.

The CNN-Module used in our experiment is shown in Fig 4. As the CNN module, we adopted the Encoder of DeepLab v3+ [113] that we utilize for pre-training, as described in the following section. Since the output of DeepLab v3+ Encoder is a feature map, we added

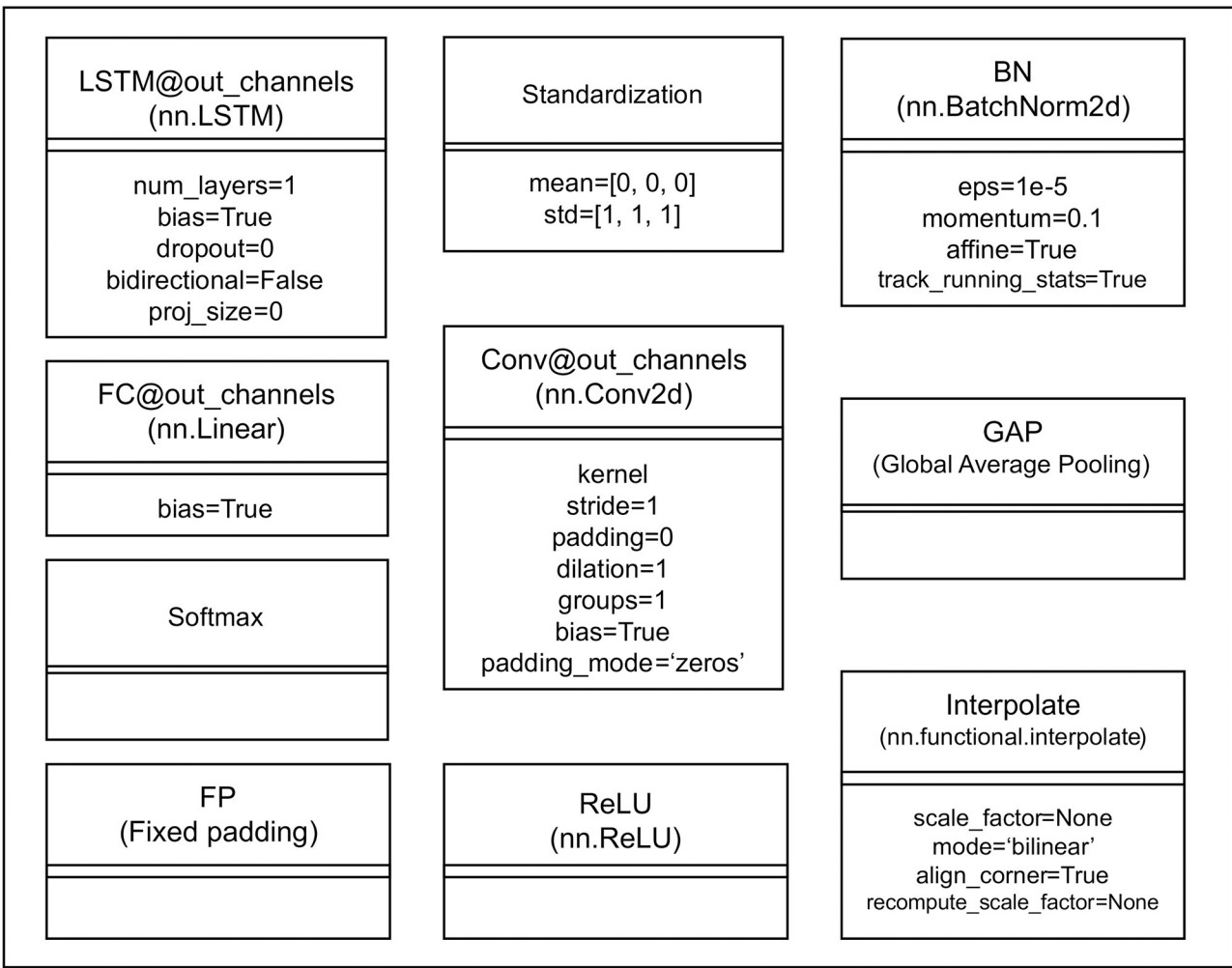

**Fig 3. Class diagram of the layers used in the model diagrams in Figs 2 and 4.** This figure describes the hyper-parameters that can be set for each class and their default values.

Vectorizer right after Encoder to convert it into a vector. Hyper-parameters for each layer were determined as follows: the hyper-parameters for Encoder of DeepLab v3+ were the values recommended in the original paper. To effectively reduce the feature resolution before performing GAP, we added three additional layers of CNNs with stride = 2 as Vectorizer. The hyper-parameters of these CNNs were determined through grid search by comparing validation scores. The mean and standard deviation parameters used in the standardization function were calculated from the entire image of training set. Therefore, the result of the calculation, $f_t$, is a $d$-dimensional feature vector that represents the spatial information of $x_t$.

The LSTM module performs the following calculation:

$$Y_n = LSTM(F_n) \tag{2}$$

where $F_n = \{f_n, \ldots, f_{n+T}\}$ denotes a series of output feature vectors from $CNN(X_n)$, which are computed independently. The result, $Y_n = \{y_n, \ldots, y_{n+T}\}$, denotes a corresponding series of output action labels.

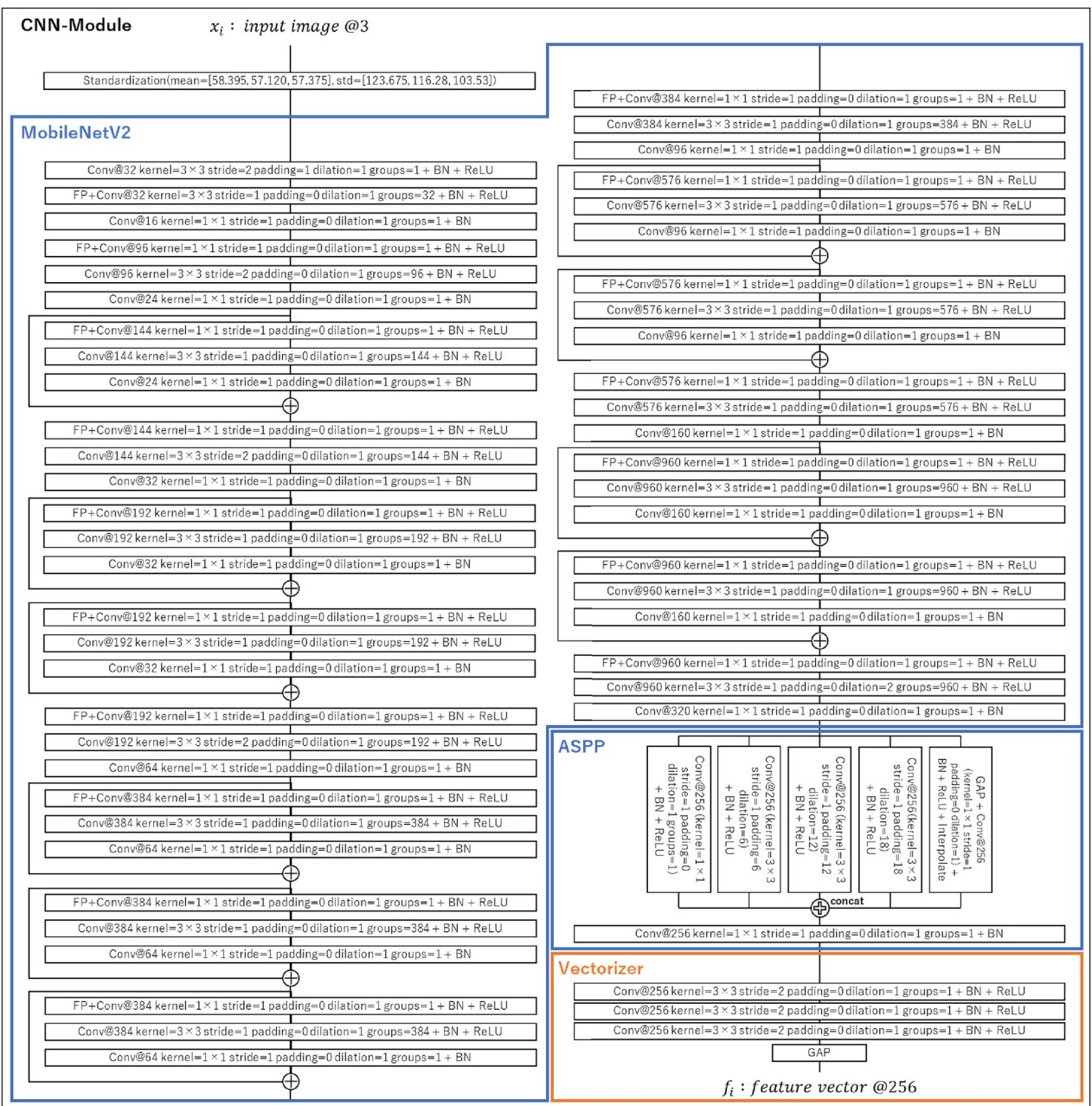

**Fig 4. Our CNN-Module architecture.** All layers of blue boxes are the encoder of DeepLab v3+ [113] (MobileNetV2 [115]-backbone). Vectorizer is a CNN+GAP layer that we added to convert feature maps to vectors. Each layer component corresponds to the class diagram in Fig 3. Hyper-parameters not listed use the default values from Fig 3.

The loss function is computed as

$$Loss(Y_n) = \sum_{t=n}^{n+T} CE(gt_t, y_t) \tag{3}$$

where $gt_t$ is the ground truth at time $t$, and $CE(\cdot)$ is a cross-entropy loss function.

**Algorithm 1** Training factory segmentation pre-trained CNN-LSTM

```
Require: X_n = {x_n, x_{n+1}, ..., x_{n+T}}, gt_i ∈ GT (GT: work ground truth set)
Ensure: Y_n = {y_n, y_{n+1}, ..., y_{n+T}} (y = predicted work label)
 1: while Until Convergence do
 2:   for NUMBER_of_IMAGES / BATCH_SIZE do
 3:     L ← 0
 4:     for BATCH_SIZE do
 5:       feature_maps_n ← Encoder(X_n) # Parallel, Freeze weight
 6:       feature_vectors_n ← Vectorizer(feature_maps_n) # Parallel
 7:       Y_n ← LSTM(feature_vectors_n)
 8:       L ← L + ∑_{t=0}^{T} Loss(gt_{n+t}, y_{n+t})
 9:     end for
10:     Adam(L) then update params
11:   end for
12: end while
```

## 4.2 Factory segmentation pre-training

The factory segmentation pre-training of the CNN-Module is shown in Fig 5. The algorithm of this step is explained in Algorithm 2.

Pre-training [116] is a learning method in which models are trained previously using a different dataset than the target task uses; the use of pre-trained models can be expected to reduce over-fitting in the target task. Traditionally, large-scale general image datasets such as ImageNet [117] and MSCOCO [118] are often used as pre-training datasets in neural net models for image processing. The CNN-Module used in the baseline model in this study was also pre-trained on ImageNet. However, pre-training with general image datasets such as ImageNet is only useful for capturing general features in images and is not suitable for extracting task-specific features. [119] successfully improved model classification performance in a study of pattern classification of interstitial lung diseases from lung CT images by pre-training the model on general texture datasets in addition to lung CT images to address the lack of training data. Thus, a more domain-specific form of pre-training may be more effective than pre-training on a general dataset alone for applied tasks for which training data are limited.

There are heuristics for identifying the important features in a frame image (e.g., position of a worker's body part, position and state of a machine tool) for work detection in practical

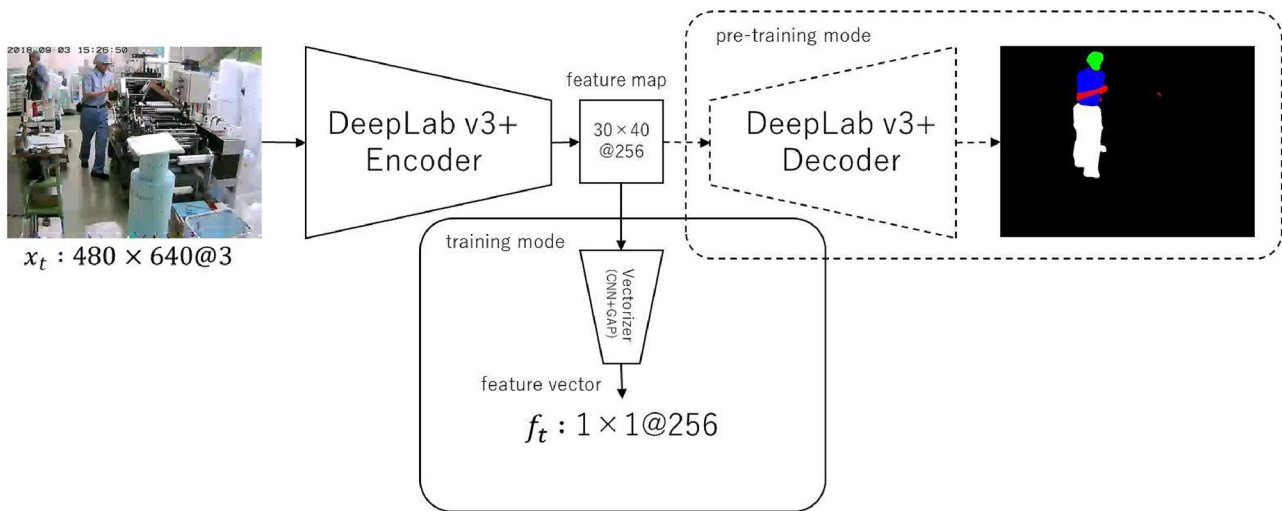

**Fig 5. Our segmentation heuristics factory pre-training architecture.**

applications. Because the CNN-Module used in this study is only an image feature extractor (the so-called encoder mechanism), the CNN-Module is generally pre-trained for an image recognition task that uses only the encoder. However, the image recognition task only determines whether or not there is a target in the image, which makes it difficult to utilize the heuristics. Therefore, in this study, we constructed an encoder–decoder model by adding a decoder mechanism after the encoder, which is not used in the work detection model but only during pre-training, so that the object segmentation task can be learned. Because segmentation performs pixel-level object classification, it is possible to extract more highly representative features. In this study, we trained the model for the segmentation of the worker's body part, which is the most common heuristic in factory work detection. More advanced heuristics, such as the segmentation of tools and specific machine parts, can be included in applications in the real-world. It is expected that this pre-training will allow the encoder to extract more domain-specific features with an emphasis on heuristics.

We used DeepLab v3+ [113] as a segmentation model for pre-training because it is a simple encoder–decoder model consisting of a single stage and is thus naturally applicable.

We randomly sampled 337 frames from the training set of our factory video dataset so that work labels were distributed as evenly as possible. We annotated the worker's head, upper body, lower body, and arm segments in these images to create a dataset for the segmentation task. We used this dataset to train the ImageNet pre-trained DeepLab v3+ model. The dataset was holdout validated with train = 269 images and test = 68 images, and the evaluation result on the test set was mIoU = 0.86.

After the above pre-training, learning is transferred to the task of work detection. For this, we remove the decoder in the segmentation-pre-trained model because the decoder is not needed for work detection. In addition, because the output of the encoder is a feature map, it is transformed into a vector by a CNN consisting of three layers and GAP (Global Average Pooling) [120].

**Algorithm 2** Pretraining Encoder with factory segmentation

```
Require: x_i ∈ X, gt_i ∈ GT (GT: segmentation ground truth set)
Ensure: y_i ∈ Y (Y: predicted segmentation set)
 1: while Until Convergence do
 2:   for NUMBER_of_IMAGES/ BATCH_SIZE do
 3:     L ← 0
 4:     for BATCH_SIZE do
 5:       feature_map_i ← Encoder(x_i)
 6:       y_i ← Decoder(feature_map_i)
 7:       L ← L + Loss(gt_i, y_i)
 8:     end for
 9:     Adam(L)
10:   end for
 8: end while
```

## 5 Experiment

To ascertain the effect of the segmentation task pre-training on work detection performance, we compared the work detection performance of two methods: the CNN-LSTM model (baseline), in which the CNN-Module was pre-trained with ImageNet only, and the CNN-LSTM model (ours), which was pre-trained with the segmentation task.

In the first section, we describe the details of our dataset. Then, we describe the experiment implementation details and the results for the two methods.

## 5.1 Dataset

Here, we provide details about our factory operations video dataset. The videos were filmed using a fixed camera for three days at a specific work site in a printing factory, with a data size of 10 h per day, i.e., 30 h for the entire dataset. The dataset is annotated by the second with 12 labels according to the work content. Because this dataset has a time-series feature, it is appropriate to divide it into the training, validation, and test sets along the time series. Thus, we divided the dataset such that the training set consists of all of day 1 and the morning of day 2, the validation set consists of the afternoon of day 2, and the test set consists of all of day 3.

Table 4 shows the distribution of the labels in the dataset. It can be seen that the frequency of class occurrences is biased. This is a common situation in real-world applications of the work detection model, and it is important to train the model so that the tasks that appear less frequently can be correctly inferred.

## 4.2 Implementation details

In our experiments, we sampled a fixed length $T$ at a sampling rate $FPS$ from each frame sequence as the input. We set $T$ to 100 frames and $FPS$ to 1 fps. Hence, the frame sequence we utilized had a time length of 100 s. The output channel of the CNN-Module was set to 256. During training, we used the Adam optimizer [121] to optimize the network. The initial learning rate was set to $10^{-3}$. The batch size was 128.

Table 5 shows the equipment hardware and software infrastructure details of our experiment. Computational resource of AI Bridging Cloud Infrastructure (ABCI) provided by National Institute of Advanced Industrial Science and Technology (AIST) was used.

## 5.3 Results and comparisons

In this section, we compare the results from our proposed segmentation heuristics pre-training method with those of the ImageNet-pre-trained model.

Fig 6 shows a loss curve of our factory-segmentation-pre-trained CNN-LSTM. For prediction, we used the weights of the epochs with the highest class accuracy in the validation set. In this experiment, the best validation class accuracy was achieved at 9 epochs (training took 21 minutes), so the weights at 9 epochs were used for the experiment. In the loss curve, the validation loss and training loss both show a decreasing trend, and there is no significant overfitting.

**Table 4. Distribution of labels in our factory video dataset.**

| Class ID | Label | Proportion [%] |
|---|---|---|
| Class 1 | Chores | 7.66 |
| Class 2 | Paperwork | 6.42 |
| Class 3 | Cylinder preparation | 11.26 |
| Class 4 | Material preparation | 8.96 |
| Class 5 | Ink roller adjustment | 3.35 |
| Class 6 | Winding preparation (top) | 3.73 |
| Class 7 | Winding preparation (bottom) | 3.15 |
| Class 8 | Creation of sample | 9.11 |
| Class 9 | Ink adjustment | 1.75 |
| Class 10 | Workbench | 0.85 |
| Class 11 | Product check | 11.00 |
| Class 12 | Nonhuman work | 32.75 |

**Table 5. The equipment hardware and software infrastructure details.**

| Hardware | Product | Spec |
|---|---|---|
| Camera | PLANEX CS-QR20 | 15FPS, 640×480px |
| Computer | ABCI | CPU:Intel Xeon Gold 6148 Processor |
| | | CPU Memory:360GB |
| | | GPU:NVIDIA Tesla V100 SXM2 (16GB HBM2)×4 |
| **Software** | | **version** |
| Python | | 3.6.5 |
| CUDA | | 10.1.243 |
| cudnn | | 7.6.4 |
| PyTorch | | 1.8.0 |

Although there is a gap between these two, it is considered to be within the acceptable range for learning progress.

Table 6 summarizes our experimental results. In the dataset used in this study, the number of data in each class differs considerably, and the micro average is not suitable because the influence of some classes is large. For this reason, we used the macro average as the average score in the results shown in the table.

From the table, we can see that the proposed method achieved an F1 performance of 0.6173, better than that of the baseline (ImageNet pre-training) method of 0.3267. This result shows that the baseline model is in a quite over-fitted state. In contrast, our method has a smaller performance gap between training and testing, which demonstrates that our segmentation heuristics pre-training method contributes to the reduction in over-fitting.

In addition, we performed a Welch's t-test to confirm the significance of the difference between the output of each method. As a specific procedure, we randomly divided the dataset into 100 parts, calculated the F1 score for each class in each part, and performed Welch's t-test

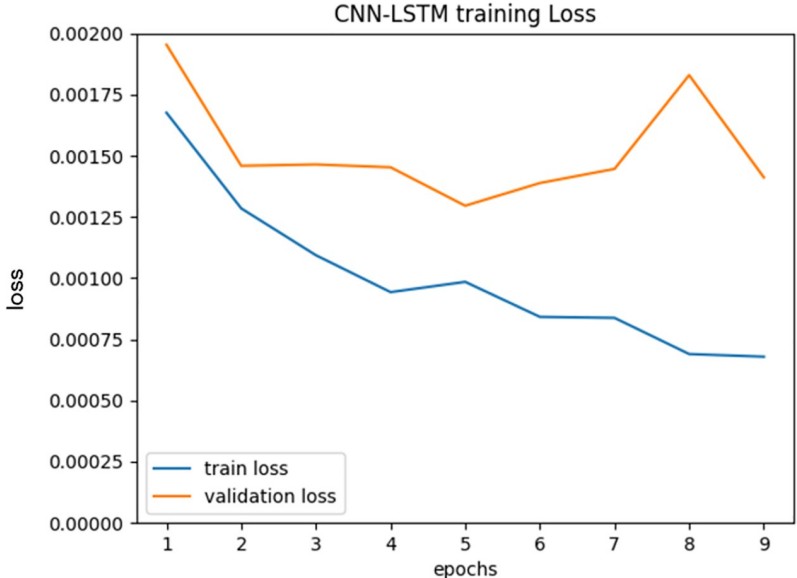

**Fig 6. Loss curve of our factory-segmentation-pre-trained CNN-LSTM training.**

**Table 6. Comparison of the two CNN-LSTM models.** The baseline is ImageNet-pre-trained CNN-LSTM; ours is the factory-segmentation-pre-trained CNN-LSTM. Scores were calculated using the macro average. The highest scores are expressed in a bold font.

| Model | | Accuracy | F1 | Recall | Precision |
|---|---|---|---|---|---|
| Baseline | training | **0.9954** | **0.9946** | **0.9947** | **0.9945** |
| | validation | 0.6364 | 0.3940 | 0.3846 | 0.4578 |
| | testing | 0.3685 | 0.3267 | 0.3528 | 0.4955 |
| Ours | training | 0.8991 | 0.8691 | 0.8614 | 0.8823 |
| | validation | **0.7152** | **0.5528** | **0.5678** | **0.6096** |
| | testing | **0.7125** | **0.6173** | **0.6221** | **0.6407** |

on each class. Consequently, the null hypothesis on all classes except class 7 F1 score can be rejected at the 1% significance level for the difference between baseline and ours.

Tables 7–9 show the recall, precision, and F1 score, respectively, for each class. In these tables, the "Diff" column was calculated as *Diff = Ours−Baseline*, which indicates the amount by which the performance of our method differed from that of the baseline method.

**Table 7. Comparison of label-by-label recall between the two LSTM-CNN models (test set).** The highest scores are expressed in a bold font.

| Class | Label | Baseline | Ours | Diff |
|---|---|---|---|---|
| Class 1 | Chores | 0.2138 | **0.5481** | +0.3343 |
| Class 2 | Paperwork | **0.8271** | 0.6530 | −0.1741 |
| Class 3 | Cylinder preparation | 0.0775 | **0.7797** | +0.7022 |
| Class 4 | Material preparation | 0.3680 | **0.8647** | +0.4967 |
| Class 5 | Ink roller adjustment | 0.2642 | **0.4786** | +0.2144 |
| Class 6 | Winding preparation (top) | 0.2651 | **0.5336** | +0.2685 |
| Class 7 | Winding preparation (bottom) | 0.3839 | **0.7198** | +0.3359 |
| Class 8 | Creation of sample | 0.1891 | **0.4427** | +0.2536 |
| Class 9 | Ink adjustment | 0.1641 | **0.4660** | +0.3019 |
| Class 10 | Workbench | 0.2335 | **0.3684** | +0.1349 |
| Class 11 | Product check | **0.8273** | 0.6775 | −0.1498 |
| Class 12 | Nonhuman work | 0.4204 | **0.9324** | +0.5120 |

**Table 8. Comparison of label-by-label precision between the two LSTM-CNN models (test set).** The highest scores are expressed in a bold font.

| Class | Label | Baseline | Ours | Diff |
|---|---|---|---|---|
| Class 1 | Chores | 0.1856 | **0.3539** | +0.1683 |
| Class 2 | Paperwork | 0.1923 | **0.7523** | +0.5600 |
| Class 3 | Cylinder preparation | **0.8037** | 0.7922 | −0.0115 |
| Class 4 | Material preparation | **0.9007** | 0.8849 | −0.0158 |
| Class 5 | Ink roller adjustment | 0.5671 | **0.8591** | +0.2920 |
| Class 6 | Winding preparation (top) | 0.3986 | **0.5883** | +0.1897 |
| Class 7 | Winding preparation (bottom) | **0.4836** | 0.3199 | −0.1637 |
| Class 8 | Creation of sample | 0.3496 | **0.5564** | +0.2069 |
| Class 9 | Ink adjustment | **0.7222** | 0.4343 | −0.2879 |
| Class 10 | Workbench | 0.0219 | **0.4468** | +0.4249 |
| Class 11 | Product check | 0.3584 | **0.7305** | +0.3721 |
| Class 12 | Nonhuman work | 0.9623 | **0.9694** | +0.0071 |

**Table 9. Comparison of label-by-label F1 score between the two LSTM-CNN models (test set).** The highest scores are expressed in a bold font.

| Class | Label | Baseline | Ours | Diff |
|-------|-------|----------|------|------|
| Class 1 | Chores | 0.1987 | **0.4301** | +0.2314 |
| Class 2 | Paperwork | 0.3120 | **0.6991** | +0.3871 |
| Class 3 | Cylinder preparation | 0.1414 | **0.7859** | +0.6445 |
| Class 4 | Material preparation | 0.5225 | **0.8747** | +0.3522 |
| Class 5 | Ink roller adjustment | 0.3605 | **0.6148** | +0.2543 |
| Class 6 | Winding preparation (top) | 0.3184 | **0.5596** | +0.2412 |
| Class 7 | Winding preparation (bottom) | 0.4281 | **0.4430** | +0.0149 |
| Class 8 | Creation of sample | 0.2454 | **0.4931** | +0.2476 |
| Class 9 | Ink adjustment | 0.2675 | **0.4496** | +0.1821 |
| Class 10 | Workbench | 0.0400 | **0.4038** | +0.3638 |
| Class 11 | Product check | 0.5002 | **0.7030** | +0.2028 |
| Class 12 | Nonhuman work | 0.5851 | **0.9505** | +0.3654 |

Our method improved F1 in all classes. The most notable improvement was in class 3, where F1 was 0.7859, an increase of 0.6445 points from 0.1414. Class 7, which showed the least improvement, still improved by 0.0149 points, and the other classes generally improved by 0.18 to 0.4 points. Class 3, which showed the greatest improvement in F1, is the cylinder preparation process; this work content is similar to that of classes 2 and 10 and requires a detailed analysis of the worker's behavior. Because our method significantly improved the detection accuracy of these work activities, we believe that our proposed factory segmentation pre-training method is particularly effective for the detailed analysis of these work activities.

Besides, the occurrence rate of classes 9 and 10 in the dataset is extremely low (1.75% for class 9 and 0.85% for class 10), and it is difficult to extract features from the work content dataset alone, as the baseline model showed. However, our method improves F1 scores of classes 9 and 10 by 0.2036 and 0.3693 points, respectively. This suggests that pre-training of heuristics can generalize the model even for tasks with a low frequency of occurrence.

## 6 Discussion

### 6.1 Analysis of model architecture

In the dataset we used, the training set and the test set have different work days, and the worker's environment can vary throughout the dataset. These aspects reflect the reality of the situation, and there would be considerable academic value in being able to generalize the model for application to a setting other than that of training.

As the baseline model uses ImageNet for pre-training, there is a high degree of flexibility in the worker's movement in the training of the task detection model. In our factory segmentation pre-training, by contrast, the heuristics are included in the pre-training so that the direction of learning can be specified at the time the work detection model is trained. For example, we pre-trained for the segmentation of the worker's body part, and this provided the direction during the training of the work detection model that the worker's pose be considered. If necessary, by pre-training for other heuristics such as segmentation of work tools or materials, we can adjust the direction of learning to focus on these movements.

As described in the previous section, our method significantly reduced over-fitting. In cases such as the present one, in which the quantity of data is limited, we believe that giving the model a heuristic direction for learning may contribute effectively to reducing over-fitting.

**Table 10. Confusion matrix of our model's result (test set).** The rows show the classes of the groundtruths (GT). The columns show the classes predicted by the model (PR). The value of each element is a count of the pairs of the GT and the PR. The diagonal line is the true positive and is expressed in a blue cell. The class numbers in the table correspond to the class IDs in Table 4.

|  |  | PR | | | | | | | | | | | |
|---|---|---|---|---|---|---|---|---|---|---|---|---|---|
|  |  | 1 | 2 | 3 | 4 | 5 | 6 | 7 | 8 | 9 | 10 | 11 | 12 |
| GT | 1 | 946 | 67 | 161 | 38 | 4 | 23 | 77 | 162 | 41 | 49 | 118 | 40 |
|  | 2 | 232 | 1227 | 253 | 31 | 6 | 27 | 9 | 37 | 4 | 7 | 38 | 8 |
|  | 3 | 160 | 12 | 3080 | 30 | 1 | 2 | 443 | 31 | 131 | 47 | 7 | 6 |
|  | 4 | 117 | 11 | 13 | 2199 | 2 | 2 | 9 | 0 | 56 | 0 | 66 | 68 |
|  | 5 | 144 | 24 | 77 | 5 | 683 | 61 | 240 | 44 | 0 | 0 | 149 | 0 |
|  | 6 | 31 | 15 | 48 | 8 | 61 | 793 | 124 | 352 | 4 | 0 | 39 | 11 |
|  | 7 | 18 | 0 | 10 | 12 | 12 | 3 | 668 | 197 | 0 | 0 | 8 | 0 |
|  | 8 | 429 | 246 | 28 | 7 | 19 | 427 | 220 | 1193 | 14 | 0 | 94 | 18 |
|  | 9 | 29 | 0 | 0 | 1 | 0 | 0 | 268 | 16 | 370 | 0 | 109 | 1 |
|  | 10 | 76 | 9 | 42 | 5 | 0 | 0 | 0 | 0 | 0 | 84 | 0 | 12 |
|  | 11 | 369 | 5 | 2 | 110 | 7 | 9 | 27 | 48 | 231 | 0 | 1754 | 27 |
|  | 12 | 122 | 15 | 174 | 39 | 0 | 1 | 3 | 64 | 1 | 1 | 19 | 6052 |

The annotation of the worker segmentation performed for pre-training was much less costly than the annotation of the work content for work detection. The work detection performance improvement of 0.2906 points for macro F1 was obtained simply by pre-training on a segmentation task with low annotation costs; this suggests that our method can improve performance whilkke keeping annotation costs low.

## 6.2 Failure cases

Here we discuss the reasons for the cases on which detection failed using this method. Table 10 shows the confusion matrix of result of our CNN-LSTM model of the test set. The rows show the classes of the groundtruths (GT). The columns show the classes predicted by the model (PR). The value of each element is a count of the pairs of the GT and the PR. The model predicted once per second.

Fig 7 shows the output feature vectors of factory segmentation pre-trained CNN-LSTM dimensionally reduced using t-SNE and mapped onto a two-dimensional graph. The 64-dimensional feature vector, which is the output of the LSTM prior to the FC+Softmax layer in Fig 2, was compressed into a 2-dimensional vector by t-SNE. The parameters of t-SNE were set to perplexity = 30 and n_iter = 1000. t-SNE dimensionality reduction was performed on the entire test set, from which 12,000 samples were randomly sampled so that all classes were evenly distributed, and a scatter plot was created.

We consider the causes for the classes having an F1 score of less than 0.5 (Table 9, namely, classes 1, 7, 8, 9, and 10. First, let us consider classes 1 and 10. These are the labels having low F1 scores with the baseline method as well and are presumably difficult to classify in the first place. As shown in Table 4, class 10 is a rare class, constituting only 0.85% of the dataset. In addition, our review of the dataset suggests that the physical movements of the workers in classes 1 and 10 are particularly diverse, making it difficult to capture their work patterns. This can be seen from the distribution of the output feature vectors in Fig 7. Class 1 (red) and class 10 (greenyellow) are widely and finely distributed around the upper right area in the figure and are mixed with other classes such as class 2 and 3.

All of the remaining classes—7, 8, and 9—are work activities performed around the rotary press. As can be seen from Table 10, these work activities are frequently misclassified for each

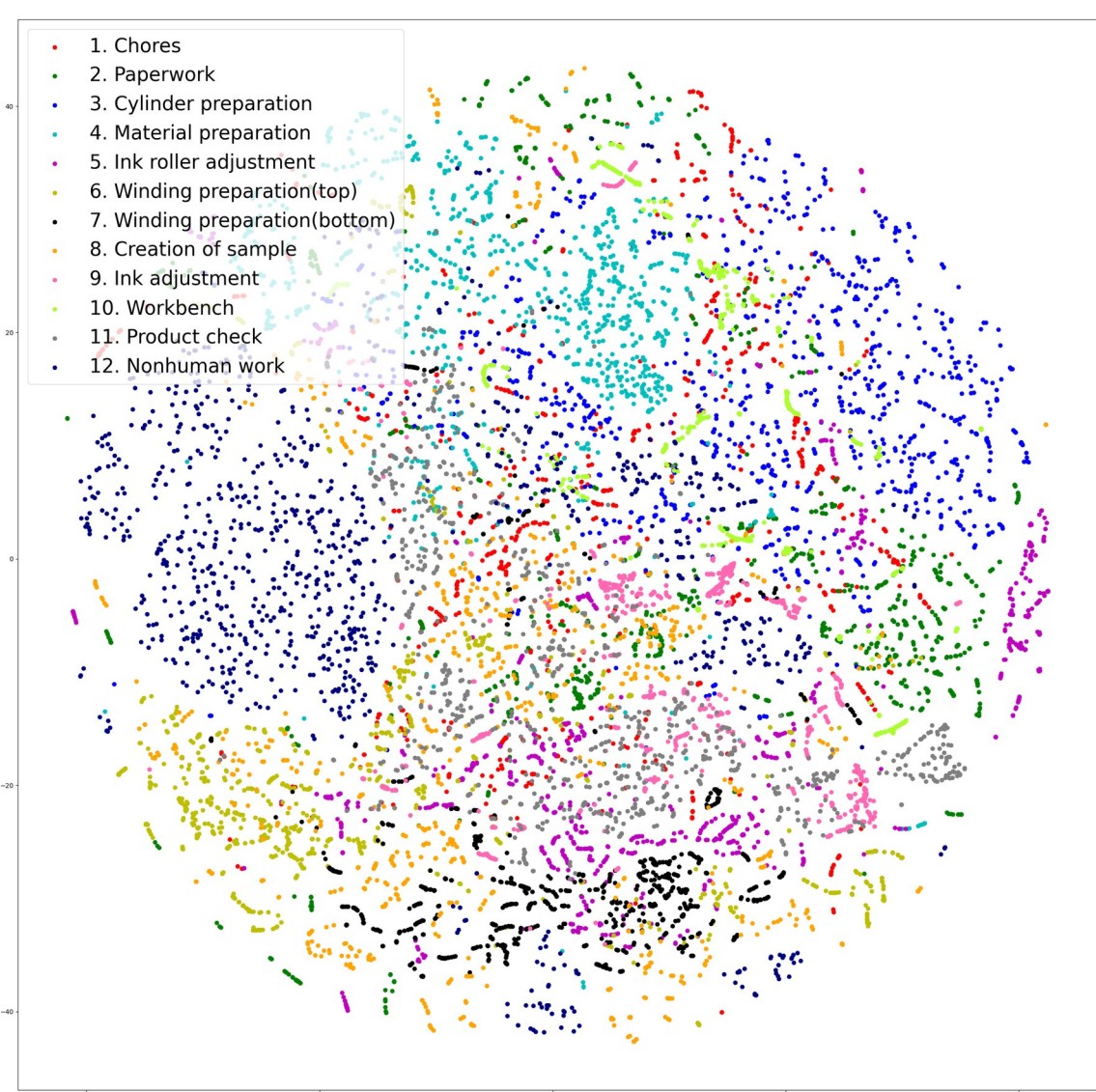

**Fig 7. t-SNE of factory segmentation pre-trained CNN-LSTM outputs. (test set).**

other. In Fig 7, these classes are widely distributed from the center to the lower left of the figure. However, they differ from classes 1 and 10 in that they form a rough clump. In many cases, even humans cannot classify these work activities correctly unless they carefully observe the operation of the rotary press and the time series of the work content. The operation of the rotary press in particular is closely related to the content of the work, and so it is quite important to take this into account.

Overall, there are sparse and dense regions in the feature space of Fig 7. Classes 3, 4, and 12 occupy a relatively large space, while the other classes are clustered in a close space. In order to deal with this, it is necessary to be able to perform feature extraction that can separate the classes that are densely clustered. For this purpose, two approaches can be considered: 1. improvement of spatial feature extraction, and 2. improvement of temporal feature extraction. For the improvement of spatial feature extraction, as mentioned above, it is possible to use

segmentation models to pre-train the machines and tools that are important for work identification. As for the improvement of temporal feature extraction, we can expect that it can be improved by using a model with higher expressive power than LSTM (Bi-LSTM, Transformer, etc.) or by using a new model that can consider the flow of work over a longer span.

## 6.3 Limitations

In this dataset, the same person performs works throughout the entire dataset. Since our method includes abstraction of the body parts of the worker by segmentation, it is considered to be robust to changes in the workers. However, the method may not be able to infer correctly when the procedures and motions of the workers change significantly due to differences in the workers. In this case, we need to add new training data.

There were cases where the working time for the same task varied from several minutes to several tens of minutes. This is thought to be because the work procedures may differ even for the same work. For example, "Product check" work is performed continuously for several tens of minutes, but it may be interrupted irregularly when defective products are found, so the work may last from a few seconds to several minutes. Since the training data contains multiple work order patterns even for the same task, the model can predict when the work order changes to some extent. However, if the order of the tasks is highly irregular, or if the tasks are completely new and unknown, the inference may not be correct. In such cases, it is necessary to add training data or review the contents of the work labels.

Our method is robust to changes in background information that is irrelevant to the task by using the segmentation of critical elements. However, since the method assumes a fixed camera, the trained model cannot be used as-is when the layout of the machines and the work environment related to the work changes. In these cases, it is considered necessary to re-collect the training data.

## 7 Conclusion

In this paper, we have proposed a factory segmentation pre-training method for video-based work detection, which is the first attempt using our proposed printing factory video dataset. This method does not interfere with the worker's work operations because it detects his or her work solely from the video. Compared with conventional methods that involve attaching sensors to workers, this system enables the detection of work activities in environments where workers have a higher degree of flexibility. The proposed pre-training method can not only learn to capture heuristics that are found important for distinguishing among similar work classes but also substantially reduce over-fitting with only a few additional segmentation annotations. Furthermore, our factory segmentation pre-training method does not result in a detection model so large that it would seriously impair the inference speed.

In addition, we proposed a printing factory video dataset, which is 30 h in length and contains annotations (12 classes) for every second. On this dataset, the proposed factory segmentation pre-training method outperforms the baseline (ImageNet pre-training) method, with an improvement of 0.2906 points in the macro F1 score.

As future research, we would like to improve our spatial and temporal feature extraction methods respectively. As for spatial features, we are interested in being able to take into account the state of machines and tools (e.g., rotation speed of the rotary press). As for temporal features, we are interested in time-series analysis methods that can extract and take into account the flow of factory operations over a longer time range.

## Author Contributions

**Data curation:** Shotaro Kataoka, Genki Iwaka, Masashi Oba.

**Formal analysis:** Shotaro Kataoka, Tetsuro Ito, Genki Iwaka.

**Investigation:** Shotaro Kataoka, Masashi Oba, Hirofumi Nonaka.

**Methodology:** Shotaro Kataoka.

**Project administration:** Shotaro Kataoka.

**Resources:** Hirofumi Nonaka.

**Software:** Shotaro Kataoka, Tetsuro Ito.

**Supervision:** Hirofumi Nonaka.

**Validation:** Shotaro Kataoka, Masashi Oba.

**Writing – original draft:** Shotaro Kataoka, Hirofumi Nonaka.

**Writing – review & editing:** Hirofumi Nonaka.

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
