## [Decision Letter · Decision Letter 0]

20 Sep 2021

PONE-D-21-26196Improving work detection by segmentation heuristics pre-training on factory operations videoPLOS ONE

Dear Dr. Kataoka,

Thank you for submitting your manuscript to PLOS ONE. After careful consideration, we feel that it has merit but does not fully meet PLOS ONE’s publication criteria as it currently stands. Therefore, we invite you to submit a revised version of the manuscript that addresses the points raised during the review process.

Specifically, the authors should better discuss the background of their study and its placement within the body of existing literature on the topic of this manuscript, improve technical presentation, present more details from experiment for replicability, and discuss the limitations of their methodology.

We look forward to receiving your revised manuscript.

Kind regards,

Robertas Damaševičius

Academic Editor

PLOS ONE

Additional Editor Comments:

The authors are invited to revise their paper following the suggestions and comments of the reviewers.

Journal Requirements:

“Unfunded studies”

Reviewers' comments:

Reviewer's Responses to Questions

**Comments to the Author**

1. Is the manuscript technically sound, and do the data support the conclusions?

Reviewer #1: Partly

Reviewer #2: Yes

2. Has the statistical analysis been performed appropriately and rigorously? 

Reviewer #1: No

Reviewer #2: Yes

3. Have the authors made all data underlying the findings in their manuscript fully available?

Reviewer #1: No

Reviewer #2: Yes

4. Is the manuscript presented in an intelligible fashion and written in standard English?

Reviewer #1: Yes

Reviewer #2: Yes

5. Review Comments to the Author

Reviewer #1: Paper aims to present an improvement (from performance perspective) of a classic CNN-LSTM model for computing spatio-temporal information. Introduction promotes "a method for considering heuristics in a model that uses an

image segmentation task as a pre-training method" as a key novelty of the paper in addition to the dataset provided at Google's server. I'm not so sure that this method is unique, as similar approaches have been done before in a different context. This requires a clarification. Authors must provide an in-depth mathematical apparatus (missing completely, only generic statements were made) and full UMLs of the implementation, including the architecture along with necessary hyper parameters of the network.

Pseudo codes provided are too shallow, failing to showcase what exactly is different from other approaches on computing spatio-temporal information. Must be expanded.

Provide loss curves (prove that your solution is not over-fitting) and tsne.

Add explanation to figure 4. What each class means.

Experimental part does not detail the equipment HW and SW infrastructure used.

Performance part (this method was supposedly much faster than accurate vs others) was not covered sufficiently.

Literature review is organized, but contains too few references. There has been a lot done in this area. Estimate around 20+ papers per section easily...

Image (video) data is available on the google drive. Yet the aggregated raw results were not provided.

Please back up this statement "manufacturing industry, the use of local computers to make inferences is often required for practicality and confidentiality reasons, necessitating a low computational cost, and so the latter, a lightweight model, needs to have improved performance". Computing costs are negligible compared to manufacturing costs and equipment.

Paper focuses on analyzing workers (live human subjects) yet provides no ethical permit.

Image (video) data is available on the google drive. Yet the aggregated raw results were not provided.

Reviewer #2: This paper presents an issue: improving work detection based on the video-analysis and on the use an image segmentation model.

1. Introduction – a very good research question and a significant problem of working time analysis. However please express the area of recognition of employee activities in the production process. So please extend your introduction this area and add the sources: https://doi.org/10.3390/sym11091151;
https://www.mdpi.com/2071-1050/13/10/5495.

2. Related work – the authors assume that background is the same throughout the video. Is the environment at the adopted position really not changing? And if so, how can this problem be solved?

3. In the video data set – the authors state, that the length of time to perform the same work consecutively varies considerably, ranging from a few seconds to tens of minutes, and tasks depend on one other over a very long period. Is the same activity performed by the same employee then? Are there no time standards for individual activities? Are the steps always performed in the same sequence?

4. Please add an explanation why DeepLab v3 was chosen as a segmentation model for pre-training - description: "it is a simple encoder – decoder model consisting of a single stage" is not sufficient.

5. How the findings can be exploited by future similar works?

6. Please add the limitations of the work.

6. PLOS authors have the option to publish the peer review history of their article (what does this mean?). If published, this will include your full peer review and any attached files.

Reviewer #1: No

Reviewer #2: No

---

## [Author Response · Author response to Decision Letter 0]

25 Nov 2021

Manuscript: PONE-D-21-26196

Response to Reviewers

Editor

Response: We did not use “Protocols.io” due to the time limitation, since the source code was newly published and the flow was defined in detail. We would like to consider it in the future works.

Reviewer 1

1. Paper aims to present an improvement (from performance perspective) of a classic CNN-LSTM model for computing spatio-temporal information. Introduction promotes "a method for considering heuristics in a model that uses an image segmentation task as a pre-training method" as a key novelty of the paper in addition to the dataset provided at Google's server. I'm not so sure that this method is unique, as similar approaches have been done before in a different context. This requires a clarification. Authors must provide an in-depth mathematical apparatus (missing completely, only generic statements were made) and full UMLs of the implementation, including the architecture along with necessary hyper parameters of the network. Pseudo codes provided are too shallow, failing to showcase what exactly is different from other approaches on computing spatio-temporal information. Must be expanded.

Response: We thank the reviewer for the recommendation. We have added the following sentences.

Uniqueness: we added a sentence to the introduction section.

“There has been no research on the application of such a method to work videos in factories. In particular, the video proposed in this study has completely new characteristics, as described in Table 2, and it would be of great academic value to confirm the effectiveness of heuristic pre-training on this video.”

Implementation details: we added or fixed figures and sentences to the method section. 

Figure2(fixed), Figure3(added)

“The CNN-Module used in our experiment is shown in Figure 3. As the CNN module, we adopted the Encoder of DeepLab v3+ [116] that we utilize for pretraining, as described in the following section. Since the output of DeepLab v3+ Encoder is a feature map, we added Vectorizer right after Encoder to convert it into a vector”

We published the CNN-LSTM code and added the following sentence to the introduction.

“The code is available at https://github.com/ShotaroKataoka/work_detection_factory_pretrain.”

2. Provide loss curves (prove that your solution is not over-fitting) and tsne.

Response: We thank the reviewer for the comment. We have added the following sentences and figures.

Loss curve: we added a sentence and a figure to the experiment section.

Figure5(added)

“Figure 5 shows a loss curve of our factory-segmentation-pre-trained CNN-LSTM. For prediction, we used the weights of the epochs with the highest class accuracy in the validation set. In this experiment, the best validation class accuracy was achieved at 9 epochs (training took 21 minutes), so we used the weights at 9 epochs for the experiment. In the loss curve, the validation loss and training loss both show a decreasing trend, and there is no significant overfitting. Although there is some gap between these two, it is considered to be within the acceptable range for learning progress.”

t-SNE: we added a sentence and a figure to the discussion section.

Figure6(added)

“Figure 6 shows the output feature vectors of factory segmentation pre-trained CNN-LSTM dimensionally reduced using t-SNE and mapped onto a two-dimensional graph. The 64-dimensional feature vector, which is the output of the LSTM prior to the FC+Softmax layer in Figure 2, was compressed into a 2-dimensional vector by t-SNE. The parameters of t-SNE were set to perplexity = 30 and n_iter = 1000. t-SNE dimensionality reduction was performed on the entire test set, from which 12,000 samples were randomly sampled so that all classes were evenly distributed, and a scatter plot was created.”

3. Add explanation to figure 4. What each class means. 

Response: We thank the reviewer for the recommendation. We have fixed figure4 and its explanation.

- Figure 4 has been changed to Table 10.

“Table 10 shows the confusion matrix of result of our CNN-LSTM model of test set. The rows show the classes of the groundtruths (GT). The columns show the classes predicted by the model (PR). The value of each element is a count of the pairs of the GT and the PR. The model predicted once per second.”

“Table 10: Confusion matrix of our model’s result (test set). The rows show the classes of the groundtruths (GT). The columns show the classes predicted by the model (PR). The value of each element is a count of the pairs of the GT and the PR. The diagonal line is the true positive and is expressed in a bold font. The class numbers in the table correspond to the class IDs in Table 4.”

4. Experimental part does not detail the equipment HW and SW infrastructure used.

Response: We thank the reviewer for the comment. We have added the Figure 5 and a sentence to the experiment section.

“Table 5 shows the equipment hardware and software infrastructure details of our experiment. Computational resource of AI Bridging Cloud Infrastructure (ABCI) provided by National Institute of Advanced Industrial Science and Technology (AIST) was used.”

5. Performance part (this method was supposedly much faster than accurate vs others) was not covered sufficiently.

Response: We thank the reviewer for the comment. We have added the following sentence to the experiment section.

“training took 21 minutes”

5. Literature review is organized, but contains too few references. There has been a lot done in this area. Estimate around 20+ papers per section easily　

Response: We thank the reviewer for the comment. As the reviewer recommended, we have added total 76 references and the detailed explanation written in red in the manuscripts. And we have also updated the “Table 2; Comparison of video datasets”.

6. Image (video) data is available on the google drive. Yet the aggregated raw results were not provided.

Response: We thank the reviewer for the comment. We have uploaded raw results to the google drive and added the following sentence to the introduction section. 

“The raw result is available along with the dataset described above.”

7. Please back up this statement "manufacturing industry, the use of local computers to make inferences is often required for practicality and confidentiality reasons, necessitating a low computational cost, and so the latter, a lightweight model, needs to have improved performance". Computing costs are negligible compared to manufacturing costs and equipment.　

Response: Thank you for your comment. It is true that the cost of a computer may be small compared to the cost of equipment. However, this system is intended for small and medium-sized companies. However, this system is designed for small and medium-sized enterprises (SMEs), which do not have ample funds and cannot easily afford to purchase computers. Therefore, it is important to have a simple model that does not require a lot of machine power.

8. Paper focuses on analyzing workers (live human subjects) yet provides no ethical permit.　

Response: We thank the reviewer for this comment. As described in the paper, the dataset videos were filmed using a fixed camera for three days at a specific work site in a printing factory (Echigo Fudagami Inc : Uenoyama 1-2-8, Ojiya, Niigata, Japan.), with a data size of 10 h per day, i.e., 30 h for the entire dataset. The data were collected after prior explanation to the company (Echigo Fudagami Inc) that the data would be used for research purposes and verbal consent was obtained from the company manager and the subject workers. Permission to conduct the research was obtained from the company manager and subject workers. Our university's ethics committee confirmed that ethical approval was waived. And no third-party ethical oversight was provided.

Reviewer 2 

1. Introduction – a very good research question and a significant problem of working time analysis. However please express the area of recognition of employee activities in the production process. So please extend your introduction this area and add the sources: https://doi.org/10.3390/sym11091151;
https://www.mdpi.com/2071-1050/13/10/5495.

Response: Thank you for your detailed response and valuable insights. We add the following sentence written in red into “Introduction” and 17 references included “https://www.mdpi.com/2071-1050/13/10/5495”.

“Recognition of employee activities in the production process is gaining attention \\cite{Bell, E.:2020}-\\cite{Graessley, S. :2019}. Among them, employee behavior recognition in the manufacturing industry is attracting increasing attention from both industry and academia \\cite{Meyers, T. D.:2019}. The importance of real-time process monitoring and analysis of employee behavior based on various sensors and other devices is increasing from both industry and academia nowadays.\\cite{White, T.:2020} - \\cite{Davis, R.:2020}”

2. Related work – the authors assume that background is the same throughout the video. Is the environment at the adopted position really not changing? And if so, how can this problem be solved?

Response: 

We thank the reviewer for the important question. Our target is a production line with many routine tasks. In this type of line, the background rarely changes, such as when the machines change. On the other hand, when the background changes, it is when the machine layout of the factory changes drastically. In such a factory where the layout changes, the work procedures and contents also change, so they are not the subject of the work analysis in this research. We will address this issue in the future.

3. In the video data set – the authors state, that the length of time to perform the same work consecutively varies considerably, ranging from a few seconds to tens of minutes, and tasks depend on one other over a very long period. Is the same activity performed by the same employee then? Are there no time standards for individual activities? Are the steps always performed in the same sequence?

Response: 

We thank the reviewer for the question. In many companies, there are no individual time standards, so they have to be identified using this kind of method. Although work procedures may change, they are included in the scope of work process analysis, and this research is able to handle them.

4. Please add an explanation why DeepLab v3 was chosen as a segmentation model for pre-training - description: "it is a simple encoder – decoder model consisting of a single stage" is not sufficient.

Response: We thank the reviewer for the valuable comment. Deep Lab V3 is one of the best performers in the field of segmentation. For example, according to the ranking of https://paperswithcode.com/, it ranks 2nd in Pascal 2012 and 1st in Anatomical Tracings of Lesions After Stroke (ATLAS). In addition, it is widely used in practical applications and has stable operation.

5. How the findings can be exploited by future similar works?

Response: As described in the paper, unlike the conventional image recognition task, the analysis of manufacturing tasks is difficult because the work time varies greatly for each task. Since this research has solved this problem, we believe that we have achieved a significant result in that we have developed a standard model for future work analysis in manufacturing and a dataset that can be used for pre-training. This research can be used to analyze various tasks in the manufacturing industry, including the detection of abnormalities in work, and will contribute to the diversification of work analysis tasks.

6. Please add the limitations of the work.

Response: We thank the reviewer for the important comment. Our proposed method cannot be used when the factory layout changes drastically. If another task is added, relearning is required.

---

## [Decision Letter · Decision Letter 1]

7 Dec 2021

PONE-D-21-26196R1Improving work detection by segmentation heuristics pre-training on factory operations videoPLOS ONE

Dear Dr. Kataoka,

Thank you for submitting your manuscript to PLOS ONE. After careful consideration, we feel that it has merit but does not fully meet PLOS ONE’s publication criteria as it currently stands. Therefore, we invite you to submit a revised version of the manuscript that addresses the points raised during the review process.

Specifically, we authors need to improve statistical analysis, discuss the limitations of the methodology and address other issues noted by the reviewers.

We look forward to receiving your revised manuscript.

Kind regards,

Robertas Damaševičius

Academic Editor

PLOS ONE

Journal Requirements:

Reviewers' comments:

Reviewer's Responses to Questions

**Comments to the Author**

1. If the authors have adequately addressed your comments raised in a previous round of review and you feel that this manuscript is now acceptable for publication, you may indicate that here to bypass the “Comments to the Author” section, enter your conflict of interest statement in the “Confidential to Editor” section, and submit your "Accept" recommendation.

Reviewer #1: All comments have been addressed

Reviewer #2: (No Response)

2. Is the manuscript technically sound, and do the data support the conclusions?

Reviewer #1: Yes

Reviewer #2: Partly

3. Has the statistical analysis been performed appropriately and rigorously? 

Reviewer #1: Yes

Reviewer #2: N/A

4. Have the authors made all data underlying the findings in their manuscript fully available?

Reviewer #1: Yes

Reviewer #2: Yes

5. Is the manuscript presented in an intelligible fashion and written in standard English?

Reviewer #1: Yes

Reviewer #2: Yes

6. Review Comments to the Author

Reviewer #1: English language still requires a rework.

Statistical analysis still requires more effort to prove the validity.

Reviewer #2: The authors did not make all corrections I had proposed. I still have comments to the improved manuscript. Why were 17 new literature references added and indicated in the recommendations not added? Moreover, the most important problem that has not been explained in the article is the incorrect assumption that "In many companies, there are no individual time standards, so they have to be identified using this kind of method". In the production process, each workstation has a different duration of activity. You stated, that "Although work procedures may change, they are included in the scope of work process analysis". So you can use the method without any requirements? This aspect requires a thorough explanation. Also, my suggestion to present the limitations of the work was as also explained laconic.

7. PLOS authors have the option to publish the peer review history of their article (what does this mean?). If published, this will include your full peer review and any attached files.

Reviewer #1: No

Reviewer #2: No

---

## [Author Response · Author response to Decision Letter 1]

24 Jan 2022

Manuscript: PONE-D-21-26196

Response to Reviewers

We have corrected the points pointed out by the reviewers. The red letters are the corrections made in the previous revision, and the blue letters are the corrections made in this revision. Underlines are grammatical corrections in the previous revision.

Reviewer 1

1. English language still requires a rework.

Response: 

 We thank the reviewer for the recommendation. We have made some grammatical corrections to our manuscript. We have underlined the corrections.

2.　Statistical analysis still requires more effort to prove the validity.

Response: 

 We thank the reviewer for the recommendation. We performed a Welch's t-test on the results. We have added the following sentences to the experiment section.

“In addition, we performed a Welch’s t-test to confirm the significance of the difference between the output of each method. As a specific procedure, we randomly divided the dataset into 100 parts, calculated the F1 score for each class in each part, and performed Welch's t-test on each class.

Consequently, the null hypothesis on all classes F1 score can be rejected at the 1\\% significance level for the difference between baseline and ours.”

Reviewer 2 

1. Why were 17 new literature references added and indicated in the recommendations not added?

Response: 

We are very sorry about it. The number of citations has been reduced by mistake. We have added one reference that was not cited previously, and one more citation ([13],[14]). Therefore, we have added 18 citations from the first edition. 

[13] Andronie, M., Lăzăroiu, G., Ștefănescu, R., Uță, C., & Dijmărescu, I. (2021). Sustainable, Smart, and Sensing Technologies for Cyber-Physical Manufacturing Systems: A Systematic Literature Review. Sustainability, 13(10), 5495.

[14] Patalas-Maliszewska, J., & Halikowski, D. (2019). A model for generating workplace procedures using a CNN-SVM architecture. Symmetry, 11(9), 1151.

The text has been changed to the following sentence.

“Recognition of employee activities in the production process is gaining 2 attention [1] [2] [3] [4] [5] [6] [7] [8] [9] [10] [11] [12]. Among them, employee behavior 3 recognition in the manufacturing industry is attracting increasing attention from both 4 industry and academia [13]. For example, to improve management and effectiveness of 5 employees ’learning processes, Patalas-Maliszewska et.al [14] used CNN-SVM to 6 extract features from video material concerning each work activity, and comparison 7 with the features of the instruction picture. The importance of real-time process 8 monitoring and analysis of employee behavior based on various sensors and other 9 devices has been increasing from both industry and academia [15] [16] [17] [18].”

2. Moreover, the most important problem that has not been explained in the article is the incorrect assumption that "In many companies, there are no individual time standards, so they have to be identified using this kind of method". In the production process, each workstation has a different duration of activity. You stated, that "Although work procedures may change, they are included in the scope of work process analysis". So you can use the method without any requirements? This aspect requires a thorough explanation. Also, my suggestion to present the limitations of the work was as also explained laconic.

Response: 

We thank the reviewer for the recommendation. In our previous reply, we misinterpreted your question and gave a wrong answer.

Each individual task has its own time standards. Throughout the dataset, the worker is the same, but there seems to be a time variance of a few minutes to a few tens of minutes when the work procedures are different. An explanation of this has been added to the manuscript with an example. 

Also, as you pointed out, the description of limitations in the paper was insufficient. We have added the following sentences to the limitations sub-section in the discussion section.

“In this dataset, the same person performs works throughout the entire dataset. Since our method includes abstraction of the body parts of the worker by segmentation, it is considered to be robust to changes in the workers. However, the method may not be able to infer correctly when the procedures and motions of the workers change significantly due to differences in the workers. In this case, we need to add new training data.

There were cases where the working time for the same task varied from several minutes to several tens of minutes. This is thought to be because the work procedures may differ even for the same work. For example, "Product check" work is performed continuously for several tens of minutes, but it may be interrupted irregularly when defective products are found, so the work may last from a few seconds to several minutes. Since the training data contains multiple work order patterns even for the same task, the model can predict when the work order changes to some extent. However, if the order of the tasks is highly irregular, or if the tasks are completely new and unknown, the inference may not be correct. In such cases, it is necessary to add training data or review the contents of the work labels.

Our method is robust to changes in background information that is irrelevant to the task by using the segmentation of critical elements. However, since the method assumes a fixed camera, the trained model cannot be used as-is when the layout of the machines and the work environment related to the work changes. In these cases, it is considered necessary to re-collect the training data.”

---

## [Decision Letter · Decision Letter 2]

31 Jan 2022

PONE-D-21-26196R2Improving work detection by segmentation heuristics pre-training on factory operations videoPLOS ONE

Dear Dr. Kataoka,

Thank you for submitting your manuscript to PLOS ONE. After careful consideration, we feel that it has merit but does not fully meet PLOS ONE’s publication criteria as it currently stands. Therefore, we invite you to submit a revised version of the manuscript that addresses the points raised during the review process.

Specifically, the authors need to address the reviewer comments seriously and provide a major revision of the manuscript according to every suggestion and comment of the reviewers.

We look forward to receiving your revised manuscript.

Kind regards,

Robertas Damaševičius

Academic Editor

PLOS ONE

Additional Editor Comments:

I urge the authors to address the reviewer comments seriously and revise the manuscript accordingly.

Reviewers' comments:

Reviewer's Responses to Questions

**Comments to the Author**

1. If the authors have adequately addressed your comments raised in a previous round of review and you feel that this manuscript is now acceptable for publication, you may indicate that here to bypass the “Comments to the Author” section, enter your conflict of interest statement in the “Confidential to Editor” section, and submit your "Accept" recommendation.

Reviewer #1: (No Response)

Reviewer #2: All comments have been addressed

2. Is the manuscript technically sound, and do the data support the conclusions?

Reviewer #1: Partly

Reviewer #2: Yes

3. Has the statistical analysis been performed appropriately and rigorously? 

Reviewer #1: Yes

Reviewer #2: Yes

4. Have the authors made all data underlying the findings in their manuscript fully available?

Reviewer #1: No

Reviewer #2: No

5. Is the manuscript presented in an intelligible fashion and written in standard English?

Reviewer #1: Yes

Reviewer #2: Yes

6. Review Comments to the Author

Reviewer #1: Literature review requires a major rework. Address each reference individually not as a group without any comment on contents of each. 1] [2] [3] [4] [5] [6] [7] [8] [9] [10] [11] [12]; 20] [21] [22] [23] [24; [25] [26] [27] [2; ] [31] [32] [33] [34] [35] [36] [37] [38] [39] [40] [41] [42] [43] [44] [45] [46] [47] [48] [49] [5;. [66] [67] [68] [69] [70] [71] [72] [73] [74] [75] [76] [77] [78] [80] [81] [82] [83]; 101] [102] [103] [104] [105] [106] [107] [108] [109] [110] [111] [112] [113] [114]. A lot of the references seem to be inserted at random to inflate the overall number and does not bring much added value. Dedicated text must be added to each reference explaining results and relation to the paper.

Figure 3 should be a proper UML.

Still not all hyper parameters are clearly listed, nor reasoning exists on how they were determined. Some are in figure 3. I suggest also adding a dedicated table.

Draw confusion matrix in color.

Figure are low res and hard to see.

Raw output data still not provided (data you draw all your results from)

Reviewer #2: The authors have significantly improved the text given in the manuscrip, but the website: https://github.com/ShotaroKataoka/work%20detection%20factory%20pretrain is not available

7. PLOS authors have the option to publish the peer review history of their article (what does this mean?). If published, this will include your full peer review and any attached files.

Reviewer #1: No

Reviewer #2: No

---

## [Author Response · Author response to Decision Letter 2]

16 Mar 2022

Reviewer 1

1. Literature review requires a major rework. Address each reference individually not as a group without any comment on contents of each. 1] [2] [3] [4] [5] [6] [7] [8] [9] [10] [11] [12]; 20] [21] [22] [23] [24; [25] [26] [27] [2; ] [31] [32] [33] [34] [35] [36] [37] [38] [39] [40] [41] [42] [43] [44] [45] [46] [47] [48] [49] [5;. [66] [67] [68] [69] [70] [71] [72] [73] [74] [75] [76] [77] [78] [80] [81] [82] [83]; 101] [102] [103] [104] [105] [106] [107] [108] [109] [110] [111] [112] [113] [114]. A lot of the references seem to be inserted at random to inflate the overall number and does not bring much added value. Dedicated text must be added to each reference explaining results and relation to the paper.

Response: 

Thank you for your suggestion. We have added comments on each reference.

2.　 Figure 3 should be a proper UML.

Response: 

We thank the reviewer for this comment. A new UML class diagram of our proposed system has been added as Figure 3. Therefore, the model diagram that was previously Figure 3 has been changed to Figure 4 which is a conceptual diagram to illustrate the layer structure of a complex model, so we have provided the simplest illustration.

3. Still not all hyper parameters are clearly listed, nor reasoning exists on how they were determined. Some are in figure 3. I suggest also adding a dedicated table.

Response: 

Thank you for your comment. A new explanation of hyperparameters has been added as Figure 3. In addition, the followings were added to "3.1 CNN-LSTM model".

“The hyper-parameters of the LSTM were determined through grid search by comparing validation scores.”

“The hyper-parameters of these CNNs were determined through grid search by comparing validation scores. The mean and standard deviation parameters used in the standardization function were calculated from the entire image of training set.”

4. Draw confusion matrix in color.

Response: 

Thank you for your comment. We have redrawn confusion matrix in color.

5. Figure are low res and hard to see. 

Response: 

Thank you for your comment. Figures 2, 4, and 5 have been changed to vector images. These have improved the resolution of the figures.

6. Raw output data still not provided (data you draw all your results from)

Response: 

Thank you for your comment. All raw data has been redistributed at the following URL. This URL is provided in the introduction. In addition, programs for score calculation, t-SNE, and T-test are now available at this URL.

https://drive.google.com/drive/folders/1yqKXnDFhRCyUmO7x2-wFZG9oLnBiB8L1?usp=sharing

Reviewer 2 

1. The website:

https://github.com/ShotaroKataoka/work%20detection%20factory%20pretrain is not available

Response: 

Thank you for your comment. It appears that the PDF link-click feature was incompatible with the underbar in the URL. Therefore, the URL was modified to be clickable.

---

## [Decision Letter · Decision Letter 3]

11 Apr 2022

Improving work detection by segmentation heuristics pre-training on factory operations video

PONE-D-21-26196R3

Dear Dr. Kataoka,

We’re pleased to inform you that your manuscript has been judged scientifically suitable for publication and will be formally accepted for publication once it meets all outstanding technical requirements.

Kind regards,

Robertas Damaševičius

Academic Editor

PLOS ONE

Additional Editor Comments (optional):

The reviewers presented a favourable opinion regarding the revised version of the article. However, the authors should check and resolve language errors.

Reviewers' comments:

Reviewer's Responses to Questions

**Comments to the Author**

1. If the authors have adequately addressed your comments raised in a previous round of review and you feel that this manuscript is now acceptable for publication, you may indicate that here to bypass the “Comments to the Author” section, enter your conflict of interest statement in the “Confidential to Editor” section, and submit your "Accept" recommendation.

Reviewer #1: All comments have been addressed

Reviewer #2: All comments have been addressed

2. Is the manuscript technically sound, and do the data support the conclusions?

Reviewer #1: Yes

Reviewer #2: Yes

3. Has the statistical analysis been performed appropriately and rigorously? 

Reviewer #1: Yes

Reviewer #2: N/A

4. Have the authors made all data underlying the findings in their manuscript fully available?

Reviewer #1: Yes

Reviewer #2: Yes

5. Is the manuscript presented in an intelligible fashion and written in standard English?

Reviewer #1: Yes

Reviewer #2: Yes

6. Review Comments to the Author

Reviewer #1: My remarks have been addressed in this iteration, therefor I have no objection in further processing of this paper. I would still recommend a thorough review of English wording and style, or a dedicated service.

Reviewer #2: The authors have improved the text given in the manuscript. I appreciate the change you made. I suggest to accept the paper.

7. PLOS authors have the option to publish the peer review history of their article (what does this mean?). If published, this will include your full peer review and any attached files.

Reviewer #1: No

Reviewer #2: No

---

## [Editor Report · Acceptance letter]

11 May 2022

PONE-D-21-26196R3 

Improving work detection by segmentation heuristics pre-training on factory operations video 

Dear Dr. Kataoka:

I'm pleased to inform you that your manuscript has been deemed suitable for publication in PLOS ONE. Congratulations! Your manuscript is now with our production department. 

Kind regards, 

on behalf of

Professor Robertas Damaševičius 

Academic Editor

PLOS ONE